# Institut Pasteur Dakar Mobile Lab: Part of the Solution to Tackle COVID Pandemic in Senegal, a Model to Be Exploited

Amary Fall [1,†], Idrissa Dieng [1,†], Cheikh Talibouya Touré [1,†], Moufid Mhamadi [1], Bacary Djilocalisse Sadio [1], Marie Henriette Dior Ndione [1], Moussa Moise Diagne [1], Mignane Ndiaye [1], Mamadou Aliou Barry [1], Yague Diaw [1], Allé Baba Dieng [2], Ndeye Maguette Diop [3], Safietou Sankhe [1], Boly Diop [2], Mamadou Ndiaye [2], Amadou Diallo [1], Mamadou Diop [1], Mamadou Dieng [3], Aurélie Cappuyns [4], Steven Pauwels [4], Babacar Gning [1], Gamou Fall [1], Manfred Weidmann [5], Cheikh Loucoubar [1], Rudi Pauwels [4], Amadou Alpha Sall [1], Ndongo Dia [1], Ousmane Faye [1] and Oumar Faye [1,*]

1   Pasteur Institute of Dakar, 36 Avenue Pasteur, Dakar P.O. Box 220, Senegal
2   Prevention Department, Ministry of Health, Dakar P.O. Box 4024, Senegal
3   Medical Region of Diourbel, Ministry of Health, Dakar P.O. Box 4024, Senegal
4   PRAESENS Foundation, Brussels, Belgium
5   Brandenburg Medical School Theodor Fontane, Fehrbelliner Str. 38, 16816 Neuruppin, Germany
*   Correspondence: oumar.faye@pasteur.sn; Tel.: +221-33-839-92-23
†   These authors contributed equally to this work.

**Abstract:** The COVID-19 pandemic required massive testing of potential patients in resource-constrained areas in Senegal. The first case of COVID-19 was reported on 2 March 2020 in Dakar city and on 10 March, the first cases were reported in Touba city, the second most populous city in Senegal. Following the scale of confirmed COVID-19 cases in Touba city, the Institut Pasteur de Dakar mobile laboratory truck (MLT) was deployed on March 13 to bring diagnostics to the point of need for better management of patient and outbreak control. The MLT deployed is a $6 \times 6$ truck equipped with an isolator for sample inactivation, a generator and batteries to ensure energy autonomy, and a molecular platform for pathogens detection. Nasal and oropharyngeal swabs were collected from suspected COVID-19 cases and sent to the MLT located at the Touba primary healthcare center. Samples were extracted and RNA amplified by real time qRT-PCR. A total of 11,693 samples were collected from 14 regions of Senegal and tested between March to August 2021. Within the samples tested, 10.6% (1240/1693) were positive for SARS-CoV-2. Furthermore, the MLT allowed the confirmation of the first cases of COVID-19 in 25 out of 79 health districts of Senegal. Thereby, the MLT deployment during the first 6 months of COVID-19 in Senegal allowed rapid processing of suspected case samples collected in Touba and other surrounding areas and, thus, significantly contributed to the outbreak response and early case management in Senegal.

**Keywords:** COVID-19; mobile laboratory; deployment; rapid diagnosis; Touba

## 1. Introduction

On 31 December 2019, the World Health Organisation country office in China was informed of the occurrence of pneumonia cases linked to an unknown pathogen. On 7 January 2020, a coronavirus-like causative agent was identified [1]. By May 2022, the ensuing COVID-19 pandemic affected 516,922,683 people and caused more than 6,256,945 deaths in more than 200 countries worldwide [2]. A multi-factorial approach was needed for COVID-19 control, with epidemiological investigation, laboratory diagnostics, and surveillance playing crucial roles in case confirmation, contact tracing, and patient care. COVID-19 diagnosis was initially performed only in laboratories and different tests to confirm COVID-19 evolved, including virus isolation by cell culture, antibody-capture enzyme-linked immunosorbent assay (ELISA) for antigen detection, reverse transcriptase polymerase chain reaction (RT-PCR), and real time RT-PCR assay for RNA detection [3]. These diagnostic

tools are limited by the availability of laboratory infrastructure in low-income settings, which prevents early and accurate diagnosis and, subsequently, treatment.

In Senegal, the first case of COVID-19 was reported on 2 March 2020 in Dakar city [4]. Following this first case, and with the rapid movement of people between regions, Touba city, an agglomeration of almost 2 million inhabitants, the second largest urban area after Dakar, reported its first case on 10 March 2020 [4]. Following the request of the Senegalese Minister of Health and the developing scale of COVID-19, the Institut Pasteur de Dakar (IPD) mobile laboratory Truck (MLT), which is a donation from the Praesens foundation, was deployed on 13 March 2020 in Touba to facilitate early detection of cases, patient management, epidemiological investigation, and contact tracing around confirmed cases. In this article, we report the activities of the first 6 months, March to August 2020, of the MLT deployment in Touba, and discuss possible implementations of the pathway that was built to face potential emergencies.

## 2. Materials and Methods

### 2.1. Organization of the Laboratory

The MLT deployed is a 6 × 6 truck equipped with a generator and batteries to ensure energy autonomy. It is also equipped with an isolator for sample inactivation, and an integrated platform for the molecular diagnosis of pathogens [5]. The workflow is detailed in Figure 1, with sample inactivation performed at the MLT using the isolator, extraction, and amplification in Touba Darou Khoudoss Healthcare Center laboratory.

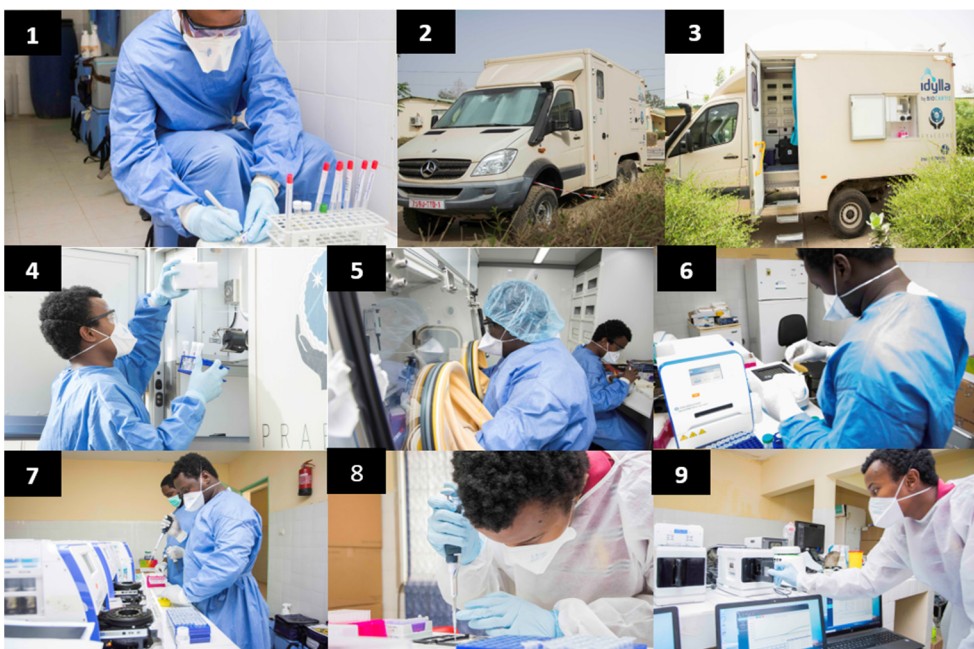

**Figure 1.** Workflow in laboratory: (**1**)—sample identification, (**2**,**3**)—MLT truck, (**4**)—sample introduction in isolator via the secured exterior sample hatch, (**5**)—sample inactivation in the isolator, (**6**)—sample lysis with the heat block, (**7**)—nucleic acid extraction, (**8**,**9**)—SARS-CoV-2 screening by RT-PCR.

### 2.2. Sample Collection

Patient samples were collected at the health facilities by medical staff (physicians, nurses) and at patient homes by field investigation teams. Nasal and oropharyngeal swabs were taken using dry swabs or transport medium, and then sent to the MLT in triple packaging. A COVID-19 investigation form containing patient identification, clinical information, and travel information was also filled out and sent to the MLT.

*2.3. Sample Handling and RNA Extraction*

Two laboratory technicians were in charge of molecular diagnostic activities, cleaning of the laboratory, and daily reporting. Suspected SARS-CoV-2 samples were introduced from the outside directly into the isolator using the airlock. Once inside the isolator, the samples tubes were inactivated using the 16TU-CV19 kit lysis buffer (Cat.No.7A133, Micobiomed, Gyeonggi-do, Korea). Nucleic acid extraction was performed using the Veri-Q PREP M16 (Micobiomed, Gyeonggi-do, Korea) device following the manufactures recommendations. Briefly, after preparation of washing solutions, the Inactivation step was performed by adding 400 μL of swab sample to 400 μL of lysis buffer based on a salt-based chaotropic agent into a 1.7 mL tube. This mix was than vigorously vortexed, the vial was externally decontaminated with aniospray disinfectant (Laboratories Anios, Lille, France), and then incubated outside the isolator in a heat block at 65 °C for 10 min. After the incubation step, 400 μL of isopropanol was added, vortexed, then loaded into a silica-gel-membrane-containing tube, then inserted into the Veri-Q PREP M16 device facilitating automated extraction on 16 glass fibre filter columns using air pressure. Pressure was applied to allow selective binding of nucleic acids to the membrane. The device then ran a sequence of alcohol-buffer-based washing steps and application of pressure (2–3 times) in order to increase purity yield. At the end of the run, the RNA was eluted in 50 μL of elution buffer then stored to 4 °C until use.

*2.4. RT-PCR Diagnostic Assays*

For the detection of SARS-CoV-2, viral RNA was subjected to real time RT-PCR using the nCOV-QS kit (Micobiomed, Gyeonggi-do, Korea). This kit is optimized for use in the Veri-Q PCR 316 cycler (Cat.No.9R501, Micobiomed Co., Ltd., Gyeonggi-do, Korea). The detection method is based on Taqman chemistry in a plastic labchip.

Briefly, the reaction master mix was prepared by adding 5 μL of buffer containing polymerase, reverse transcriptase, and stabilizer, 1 μL of primer/probe mixture nCOV PPM2 (N gene) and PPM1 (ORF1 gene), and 1 μL of internal positive control.

QRT-PCR reactions were performed in a volume of 10 μL. Briefly, 3 μL of viral RNA extract was added to 7 μL of reaction mixture, centrifuged at 3000 rpm for 2 s, and then 8 μL of the obtained mixture was loaded into each labchip channel in the order of negative control, templates, positive control. Finally, the labchip was assembled with rubbers gaskets and stoppers, and the labchip was inserted into the Veri-Q PCR 316 cycler. Real time RT-PCR was performed with reverse transcription at 55 °C/5 min, an initial denaturation at 95 °C/8 s, 45 cycles of two-step amplification at 95 °C/9 s, and an annealing phase at 56 °C/13 s. All the results were based on Ct values automatically calculated by the software. Any Ct values < 32 were considered as positive; Ct values > 32 were confirmed by testing a second patient sample.

*2.5. Sequences Generation and Analysis*

RNA was used as a template for first-stranded cDNA synthesis using the SuperScript III Reverse Transcriptase kit (Invitrogen, Thermo Fisher, Waltham, MA, USA). The double-stranded cDNA and sequencing libraries were produced using the TruSeq RNA Exome kit (Illumina, San Diego, CA, USA) previously known as TruSeq RNA. Access Library Prep Kit (Illumina, San Diego, CA, USA). Libraries were then sequenced in paired-end mode on an Illumina MiSeq (Illumina, San Diego, CA, USA), with a run of 2 × 150 bases.

Demultiplexing, removal of sequencing adapters, and generation of consensus genome were performed by using a homemade package incorporated in the IPD server. Sequences were aligned using MAFFT (9). The obtained alignment was visualized and manually curated using geneious prime 2021 (Biomatters, Auckland, New Zealand). The web-based Pangolin tool was used to assign genetic lineages.

### 2.6. Data Management and Statistical Analysis

Continuous variables were expressed as medians and range. They were compared using the Mann–Whitney U test. The Fisher test or chi-2 test were used for comparison of categorical variables such as frequencies and proportions. Odds-ratios (ORs) and 95% confidence intervals (CI) were also calculated. A $p$-value < 0.05 was considered statistically significant. The R software (R.3.0.1 version) was used to perform the statistical analyses

### 3. Results

During the six months of deployment, from 12 March to 30 August 2020, overall a total of 11,693 samples were tested in the MLT with 10.6% (1240/11,693) testing positive for SARS-CoV-2 (Table 1). These samples were collected from 55/79 health districts in 10/14 regions of Senegal (Figure 2). The majority of the samples (40.8%) were collected from the Diourbel region with a positivity rate of 64.9% (801/1240). Of these, 81.6%, 8.8%, 5.9%, and 3.5% were collected in the Touba, Diourbel, Mbacke, and Bambey health districts, respectively, and a positivity rate of 84.5% (677/801), 8.6% (69/801), 6.4% (51/801), and 0.5% (04/801) was determined, respectively.

The remaining 60.2% of samples were collected in Louga, Kaolack, Fatick, Saint-Louis, Matam, Kaffrine, Kedougou, Tamba, Dakar, Ziguinchor, Sedhiou, and Thies. During this deployment, the MLT confirmed the first cases of SARS-CoV-2 in 23/79 Senegalese health districts (Figure S1).

As for the turnaround time for results after reception, 99.5% (11,635/11,693) of the samples received during the 6 months of deployment were returned in less than 24 h, compared to the reference laboratory, which returned 40% (37,062/92,726) of the results in less than 24 h and 60% (55,664/92,726) in 24 h (Figure 3).

**Table 1.** Distribution of SARS-CoV-2 in relation to demographic characteristics and clinical signs.

| | All Patients | Sars-CoV-2 Pos. | Sars-CoV-2 Neg. | OR (95% IC) | $p$-Value |
|---|---|---|---|---|---|
| Positivity no. (%) | 11,693 | 1240 (10.6) | 10,453 (89.4) | | |
| Gender no. (%) | | | | | |
| Male | 6981 (59.7) | 743 (59.9) | 6238 (59.8) | 1.0 (0.8–1.1) | |
| Female | 4712 (40.3) | 497 (40.0) | 4215 (40.3) | | |
| Age (years) no. (%) | | | | | |
| [0–5] | 631 (5.4) | 31 (2.5) | 600 (05.7) | 0.05 (0.0–0.07) | <0.0005 |
| [6–14] | 1059 (9.1) | 53 (4.27) | 1006 (9.62) | 1.0 (0.6–1.6) | 0.933 |
| [15–50] | 7249 (62.0) | 754 (80.8) | 6495 (62.1) | 2.2 (1.5–3.3) | <0.0005 |
| 50+ | 2676 (22.9) | 396 (31.9) | 2280 (21.8) | 3.3 (2.3–4.9) | <0.0005 |
| Missing | 78 (0.7) | 6 (0.48) | 72 (0.68) | 1.0 (0.5–3.7) | 0.302 |
| Clinical signs no. (%) | | | | | |
| Fever | 2891 (24.7) | 572 (46.1) | 2319 (22.2) | 3.0 (2,6–3.3) | <0.0005 |
| Cough | 2231 (19.1) | 445 (35.9) | 1786 (17.1) | 2.7 (2.3–3.0) | <0.0005 |
| Sore throat | 899 (7.7) | 111 (08.9) | 0788 (07.5) | 1.2 (0.9–1,4) | 0.08 |
| Headache | 1773 (15.2) | 378 (30.4) | 1395 (13.3) | 2.8 (2,4–3.2) | <0.0005 |
| Myalgia | 400 (3.4) | 112 (09.0) | 288 (02.8) | 3.5 (2.7–4.3 | <0.0005 |
| Dyspnea | 547 (4.7) | 086 (06.9) | 461 (04.4) | 1.6 (1.2–2.0) | 0.0001 |
| Rhinorrhea | 612 (5.2) | 117 (09.4) | 495 (04.7) | 2.0 (1.6–2.5) | <0.0005 |
| Ageusia | 114 (1.0) | 037 (03.0) | 077 (0.7) | 4.1 (2.7–6.1) | <0.0005 |
| Anosmia | 229 (2.0) | 062 (05.0) | 167 (1.6) | 3.2 (2.4–4.3) | <0.0005 |
| Diarrhea | 077 (0.7) | 010 (00.8) | 067 (00.6) | 1.2 (0.6–2.4) | 0.2 |
| Vomiting | 080 (0.7) | 009 (00.7) | 071 (00.7) | 1.0 (0.4–2.14) | 0.4 |

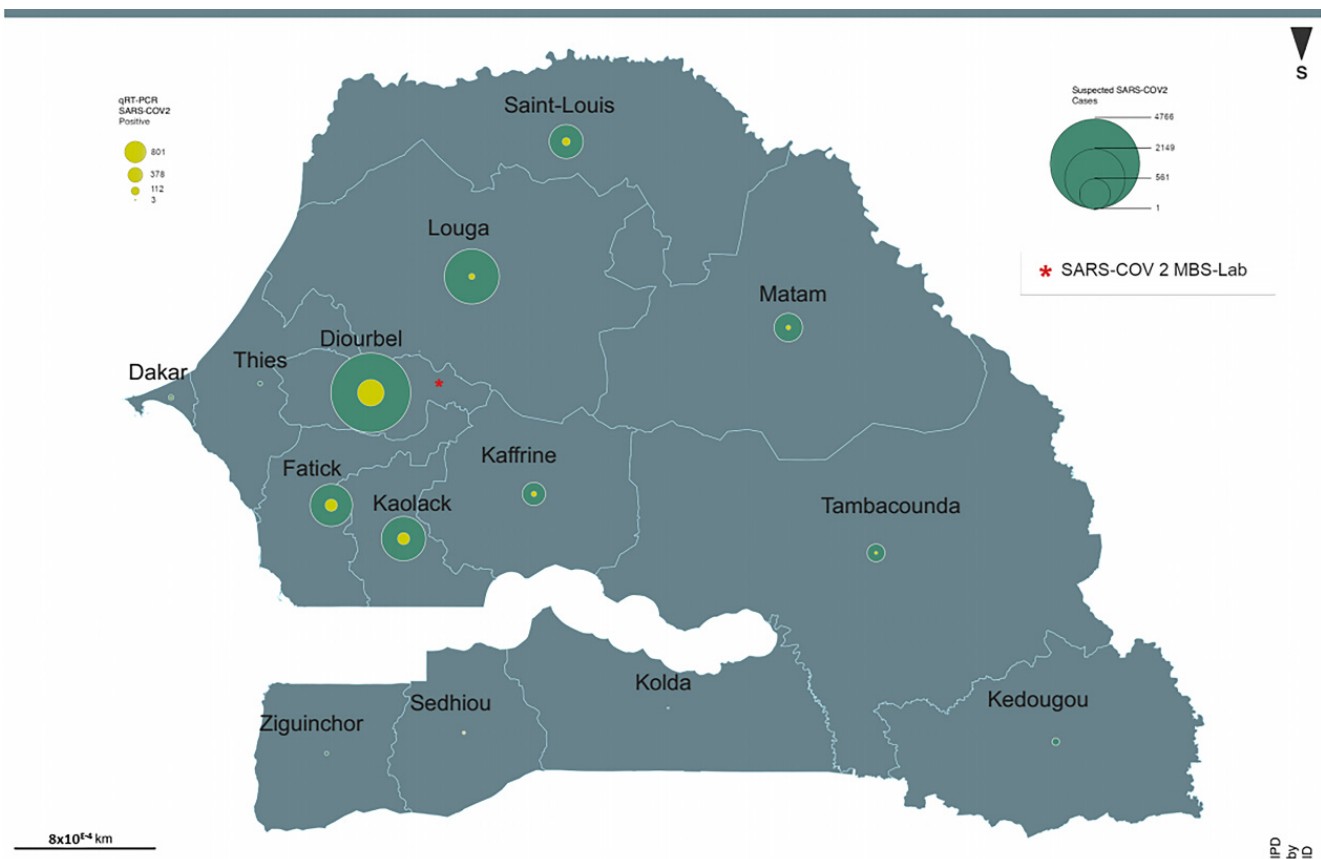

**Figure 2.** Origins of samples tested by MLT. In green the number of SARS-CoV-2 suspected cases and in yellow the number of qRT-PCR SARS-CoV-2 positives.

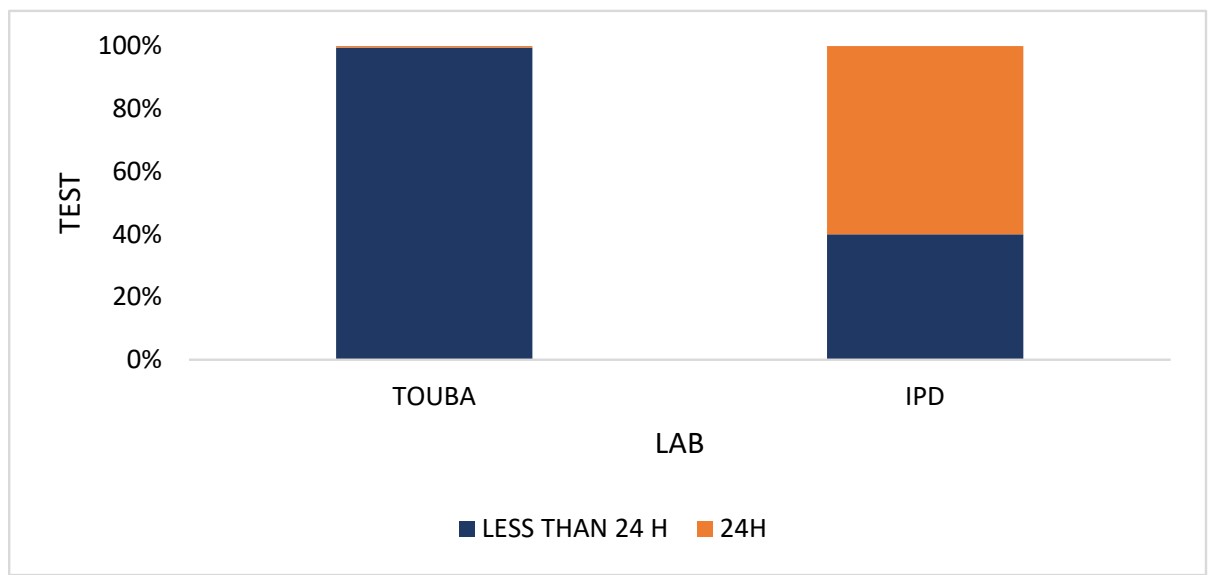

**Figure 3.** Turnaround time of results after samples reception at the MLT implemented in Touba and Institut Pasteur Dakar (IPD) laboratory.

The age of patients ranged from 3 days to 100 years, with a mean age of 34.8 $\pm$ 1.6 years and a median age of 33 years. The male/female ratio was 1.5 with 59.7% males, and no significant difference was found. Adults aged 15 to 50 were mostly represented in our study population (62.0%) with 80.8% of positives vs. 62.1% of negatives (OR: 2.2, 95% CI: 1.5–3.3). The elderly had a representation of 22.9% with 31.9% of positives vs. 21.8% of

negatives (OR: 3.3, 95% CI: 2.3–4.9). The pediatric population represented 14.5% of all our patients with 2.5% of positives vs. 5.7% of negatives (OR: 0.05, 95% CI: 0.01–0.07) (Table 1).

Regarding clinical symptoms, fever (46.1% vs. 22.2%) (OR: 3.0, 95% CI: 2.6–3.3), cough (35.9% vs. 17.1%) (OR: 2.7, 95% CI: 2.3–3.0), and headache (30.4% vs. 13.3%) (OR: 2.8, 95% CI: 2.4–3.2) were significantly found in SARS-CoV-2 positive patients. Other signs such as ageusia (3.0% vs. 0.73%) (OR: 4.1, 95% CI: 2.7–6.1) and anosmia (5.0% vs. 1.6%) (OR: 3.2, 95% CI: 2.4–4.3), often encountered in respiratory infections, were also significantly present in positive cases (Table 1).

The circulation profile shows a gradual increase in cases over the months before reaching the peak in June with 346 positive cases, then slowly decreasing. However, the peak of suspected cases was observed in May, with 4047 samples tested (Figure 4).

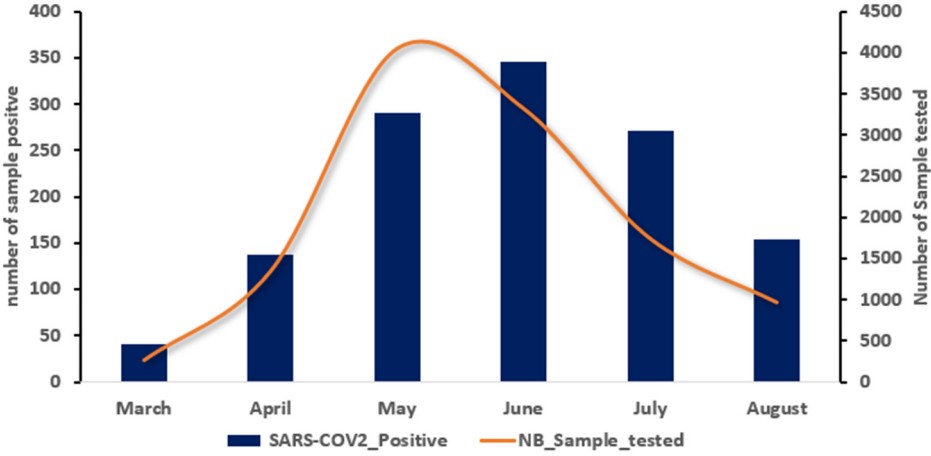

**Figure 4.** Profile of circulation of SARS-CoV-2 after six months of surveillance.

Among 27 SARS-CoV-2 genome sequences determined, seven lineages were detected (Figure S2) during the study period, the majority represented by B1 (66.67%), B.1.160 (11.11%), and A (7.40%). The other lineages, including B.1.222, B.1.247, B.1.1.293, and B.1.415, were detected with rates of 3.7% each.

Except for the region of Fatick, the B.1 lineage was detected in all the regions sampled (Figure 5). More than half of the lineages (five of seven, lineages: A, B.1, B.1.1.293, B.1.160, B.1.247) were detected in Touba health district samples, followed by Diourbel (two of seven, lineages: B.1, B.1.222). Only one lineage each was detected in the samples collected in the Fatick (B.1.415), Louga (B.1), and Saint Louis (B.1) regions.

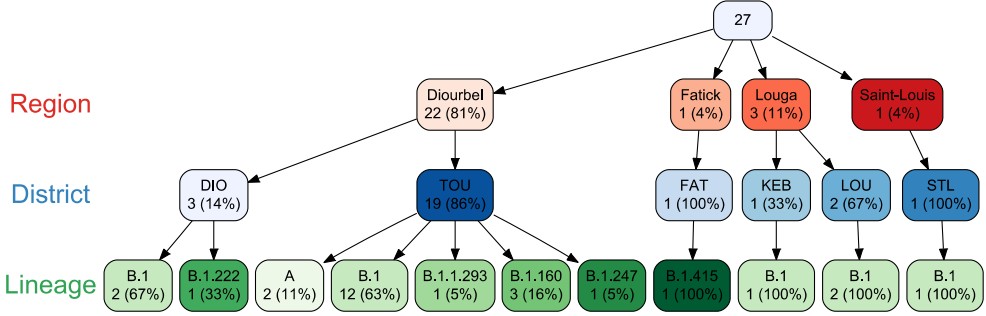

**Figure 5.** Geographical repartition of lineages.

## 4. Discussion

This study reports the results of a 6 month MLT COVID-19 response deployment to Touba city, in Senegal. A combination of several factors, including the rapidly increasing number of suspected and confirmed cases, the high population density, and the high influx of travelers to the sacred city, prompted the deployment of the MLT.

Overall, 11.693 samples were tested. The positivity rate of 10.6% was higher than the overall rate of 8.9% for Senegal during the same period [6].

The samples tested during the deployment of the MLT were collected from several health districts and regions surrounding Touba city. Therefore, the deployment of the MLT in Touba improved testing accessibility for several remote areas to quickly send COVID-19 samples to the reference laboratory located in Dakar [7], and efficient management of patients and their contacts. This also could explain the difference in turnaround time, demonstrating the added value of the unique selling point of the MLT. Its proximity at point of need combined with its autonomy (using its own LIMS in combination with a perfectly stable energy flow and mobile phone communication capabilities) facilitated quick turnaround times, and achieved results in less than 24 h in over 99% of samples tested. It would be interesting to perform further comparative analyses to determine the differences in efficiency between the reference laboratory and the MLT.

From a clinical point of view, the most common symptoms observed were fever followed by cough and headache, as widely reported for COVID-19 patients [8,9]. Regarding the age distribution, the adult and elderly patient cohorts were more affected compared to children. This confirms several studies, describing fewer severe cases in younger populations [10,11], and may be due to a lower density and affinity, as well as a different distribution of ACE-2 receptors on cells in children [12].

The infection peak during this MLT deployment occurred in June, as predicted previously [13]. The subsequent drop in confirmed cases of COVID-19 appeared to be linked to the increasing temperature due to the rainy season, which begins at the end of the second quarter of the year in Senegal. Indeed, studies show that high temperature combined with high humidity are not favorable to the transmission of COVID-19 [14,15].

The SARS-CoV-2 sequence derived of samples of the first case of COVID-19 reported in Senegal belonged to the B.1 lineage, which also was the most prevalent lineage (18/27) found in the samples tested at the MLT. Lineage B.1 was the most dominant in various African countries between March and August 2020 [16–18], and northern Italy in 2020 [19]. The first case of COVID-19 in Touba was indeed a Senegalese citizen who had traveled from Italy (MHSA, 2020b). Moreover, two European lineages of concern were detected in the samples sent to the MLT. The first of these lineages, B.1.160, called Marseille 4, was detected in samples collected in Touba, and this lineage was associated with more severe outcomes in Marseille (France) [20]. The second lineage, B.1.222, was detected in samples collected in Diourbel. This particular lineage was rapidly and widely transmitted in Scotland [21]. Further studies are needed for more insight into the dynamics of the circulation of the virus lineages.

The control of emerging diseases including early diagnostics for management of patients, investigation of cases, and prevention are very important, but hard to provide to areas with poor or non-existing health infrastructure. The IPD MLT provided high-complexity infectious disease testing to Touba during the COVID-19 pandemic. Indeed, test results that often took up to several days were available to clinicians in just 24 h, thus, reducing the cost and time involved to have couriers transporting specimens to the reference laboratory. This MLT is a solution as a rapid, real-time laboratory service for rural and remote areas without electrical power, and could also be used in other places including international/domestic airports and borders and mass gatherings to provide valuable infection surveillance. The polyvalence of the platform and its wide range of equipment makes this mobile lab fit both for epidemiological intervention at the point of prevalence, but also for surveillance and primary care support, and, as such, can be considered as diagnostic capacity building in the long-term.

## 5. Conclusions

The IPD MLT deployed in Touba during the COVID-19 response was crucial for rapid epidemic management in this region. It offers rapidly deployable technology for effective

field diagnostics capabilities in outbreak or surveillance settings. This kind of platform can be used for many other epidemics to offer a high level of support to health authorities.

**Supplementary Materials:** The following supporting information can be downloaded at: https://www.mdpi.com/article/10.3390/covid2100108/s1, Figure S1: Timeline of first case of health district detected by MLT; Figure S2: Lineages sequenced among 27 positives cases.

**Author Contributions:** Conceptualization, A.F., I.D., C.T.T., M.M., B.D.S., M.H.D.N., M.M.D., M.N. (Mignane Ndiaye), Y.D. and O.F. (Oumar Faye); data curation, A.F., I.D., C.T.T., M.M., M.M.D., A.D. and M.D. (Mamadou Diop); formal analysis, A.F., I.D., C.T.T., M.M., B.D.S., M.H.D.N., M.M.D., M.N. (Mignane Ndiaye), S.S., A.D. and M.D. (Mamadou Diop); investigation, M.A.B., A.B.D., N.M.D., B.D., M.N. (Mamadou Ndiaye) and M.D. (Mamadou Dieng); methodology, A.F., I.D., C.T.T., M.M., B.D.S., M.H.D.N., M.M.D., M.N. (Mignane Ndiaye), Y.D., S.S. and O.F. (Oumar Faye); resources, S.P., A.C., R.P., A.A.S. and O.F. (Ousmane Faye); supervision, G.F., C.L., R.P., A.A.S., N.D., O.F. (Ousmane Faye) and O.F. (Oumar Faye); validation, O.F. (Oumar Faye); writing—original draft, A.F., I.D., C.T.T., M.M. and O.F. (Oumar Faye); writing—review and editing, A.F., I.D., C.T.T., M.M., M.A.B., A.D., N.M.D., B.D., M.N. (Mamadou Ndiaye), M.D. (Mamadou Diop), A.B.D., M.D. (Mamadou Dieng), B.G., G.F., M.W., C.L., R.P., A.A.S., N.D., O.F. (Ousmane Faye) and O.F. (Oumar Faye). All authors have read and agreed to the published version of the manuscript.

**Funding:** This research received no special funding.

**Institutional Review Board Statement:** Not applicable.

**Informed Consent Statement:** Not applicable.

**Data Availability Statement:** Not applicable.

**Acknowledgments:** We would like to acknowledge the Ministry of Health for its support and all the hospital and health structures workers.

**Conflicts of Interest:** The authors declare no conflict of interest.

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
