# Peer review of "Institut Pasteur Dakar Mobile Lab: Part of the Solution to Tackle COVID Pandemic in Senegal, a Model to Be Exploited"

_covid, doi:10.3390/covid2100108_

Round 1

Reviewer 1 Report

The authors present the outcome of MLT deployment in Touba City in Senegal during the COVID 19 period. 

I believe that the manuscript describes nicely  the usefulness of mobile lab to monitor outbreaks and emergency situations. I do feel that the report could be strengthened by  attention to recommended remarks :   

Sentences to  rewrite for clear message delivery:

Line 45-46 . Please rewrite the sentence for more clarity.

Line 56-57: The sentence is incomplete

I noticed that  during the workflow details, the first step of sample identification was manually and any electronic setup  was detailed nor even how data are communicated to heath authorities or patient?

Line83-84 : Which kind of investigation form were used ? is the same for tracing?

Line 91: Sentence not complete!

Line 115 what are the cut off “Ct values” used for decision making?

Line 155-158: Do we have to draw the conclusion that MLT provide data rapid than referenced laboratory? if yes! how do you explain the difference in the number of samples treated ? what about if you compare the same number of samples ? does MLT stay performant in data delivery?

Line 181-183: You're talking about 8lineages but I saw only 7? am i right ?

Line 241-242 : Sentence not well written and pay attention to  typos and grammatical words!

Best regards

Author Response

Dear reviewer, 

Thank you for help us to improve our study

Reviewer 2 Report

Dear Editor, thank you for giving me the opportunity to revise this paper. It addresses the issue of Sars-Cov-2 diagnosis at the beginning of the pandemic, in an African context. It also offers important information on the outbreak response in that region. I read the article with interest and, probably, thanks to the efforts of editors, reviewers, and authors, the final result will be a well-written paper with potential several citations.

For this purpose, the text must necessarily be updated. I would suggest adding a final paragraph (before conclusions) in which the authors summarize their extraordinary work and look forward. What are the prospects for health surveillance? Can be the developed model applied for other emergencies? Brief sentences and notes could also be included in the abstract and perhaps also in the title (e.g., by adding ... "a model to be exploited"). The added value of publications on healthcare pathways is the possibility of identifying perspectives for future scenarios. To my knowledge, there are several surveillance programs active in Senegal. doi: 10.1016 / j.ijregi.2022.02.007, doi: 10.1136 / bmjgh-2021-007236. Thus, your paper could link this literary vein.

Moreover, did you use a similar model for vaccinations?

Abstract. The first sentence "A first case of COVID-19 was reported on December 2019 in China" is not necessary. You may start from "In Senegal ...."

Introduction

Lines 33-44. They are not necessary. After millions of articles all starting the same, this information may be superfluous ...

Lines 50-57. Probably, it would be more appropriate to write in the past tense .. "Multi-factorial approach WAS therefore needed for COVID control"

After Line 66. You could add ..." We also discuss possible implementations of the pathway that has been built to face potential emergencies..." It is an example, to stimulate readers.

Conclusion

You affirmed that "This kind of platform need to be extended in others strategic regions". What is the result of the program after more than two years? I say this because the paper seems somewhat dated. It is necessary to enter recent data to increase interest. Don't get me wrong, it is not a criticism but a suggestion. The risk is to publish an article with no prospect of citations.

Grammar check is mandatory. For example, the error is repeated several time throughout the text,A first case of COVID-19 was reported on December 2019 in China

Author Response

(The authors gave the same response as above.)

Round 2

Reviewer 2 Report

The Authors have addressed all of my concerns with the original manuscript. The revised manuscript is ready for publication